# Thermal Conductivity Enhancement Derived from Poly(Methyl Methacrylate)-Grafted Carbon Nanotubes in Poly(Methyl Methacrylate)/Polystyrene Blends

**DOI:** 10.3390/polym11081347

**Published:** 2019-08-13

**Authors:** Jaehyun Wie, Jooheon Kim

**Affiliations:** School of Chemical Engineering & Materials Science, Chung-Ang University, Seoul 156-756, Korea

**Keywords:** phase separation, polymer blends, polymer grafting

## Abstract

This paper presents a method to enhance thermal conductivity using poly(methyl methacrylate)(PMMA), polystyrene(PS) blends, and incorporation of multiwalled carbon nanotubes (MWCNTs). MWCNTs are selectively localized in PMMA phase to improve conductive properties. In addition, Surface of MWCNTs was treated with PMMA to enhance affinity between matrix and filler. PMMA grafting helps filler localization on matrix phase. Composites using two polymers enhanced thermal conductivity by ~11% compared with composites using only PS or PMMA. Also, PMMA grafting on the surface of MWCNTs enhanced thermal conductivity by ~13% compared with samples without PMMA grafting.

## 1. Introduction

Thermal management is important for the reliability, lifespan, and performance of electronic devices. While the miniaturization and integration of electronic materials like light-emitting diodes is proceeding, thermal diffusion is becoming critical. Polymers are widely used for these electronic devices because of their processability. To make up for disadvantages of polymers, like low conductive and mechanical properties, various filler incorporated composites have been used [1,2]. Carbon nanotubes (CNTs) have been considered for broad usage for filler material because of their outstanding properties like high electrical conductivity, good mechanical properties, and high thermal conductivity. However, CNTs have drawbacks for wider application. CNTs have limited dispersibility and processability due to their flocculation in bundles because of the van der Waals interactions between single carbon nanotube. CNT bundles can be dispersed in some organic solvents by sonication, but it is hard to avoid precipitation when sonication is halted. The lack of dispersibility in universal organic solvents lowers the availability of carbon nanotubes; also, CNTs are regarded to be inert for many chemical reactions.

To overcome these drawbacks of CNT, many attempts have been introduced in prior literature [2]. Covalent surface modification of CNTs with materials including polymers is commonly used. Chemical functionalization has been noted to generate chemical covalent bonding between the multiwalled carbon nanotubes (MWCNTs) and materials that react with MWCNTs. In MWCNT/polymer composites, chemical functionalization is used to increase compatibility between MWCNT and polymer interface. Increasing compatibility between MWCNTs and polymer will decrease the interfacial energy, which enhances dispersion and homogeneity of CNTs in polymer. Kim et al. treated the surface of boron nitride with silane, which stabilizes the interfacial connections [3]. The author enhanced thermal conductivity by 30% through surface modification with silane groups. The role of silane groups is like a bridge for phonon transferring, and therefore they enhance the thermal conductivity of BN. Polymer grafting is one of the method for surface functionalization of materials. Polymer grafting may grant the additional properties of CNTs which are depending on their specific nature of the grafted polymer and it also increases the affinity between polymer grafted MWCNTs and their polymer matrices [4,5,6,7]. There are two universal way for the covalent functionalization of polymers on the surface of MWCNTs, which are named as ‘grafting to’ and ‘grafting from’ [7]. ‘Grafting to’ means the attachment of a polymer by reaction between the attached functional groups and polymer chain onto the surface of MWCNTs. ‘grafting from’ means the covalent attachment of the prepolymers on the surface of the MWCNTs and polymer chain elongation with monomers. Using this method, polymers will develop on the surface of the MWCNTs.

In general, high filler loading is also required for high heat dissipation performance of composites, but high filler loading often aggravates processability and mechanical properties. Recently, unconventional method has been used to lower filler concentration in the composites, which is called double percolation in other literature [8]. Foulger et al. fabricated EVA/HDPE/CB composites [9]. EVA (ethylene vinyl acetate) and HDPE (high density polyethylene) are immiscible with each other. CB (carbon black) was preferentially localized in the HDPE, which enhanced conductive properties above that of the individually carbon filled HDPE and EVA. The conductive filler selectively localizes in their energetically preferable phase. It can increase the concentration of conductive filler in the phase, which forms network system easily with only small amount of filler. The system has especially received attention from people who were researching electric conductivity. Recently some people have brought this system for thermal conductivity [2].

This paper is focusing on improving thermal conductivity and thermal diffusivity of composites that are using two immiscible polymers for matrix and selectively localized filler. PS and PMMA are immiscible with each other, causing phase separation [10,11]. At this situation, incorporated MWCNTs are localized in favorable polymer. This phenomenon helps enhancement of thermal conductivity. To help MWCNTs localized in the PMMA phase and enhance the dispersion state of MWCNTs, polymer grafting was performed. Finally, the thermal conductivity of composites was investigated.

## 2. Experiment Section

### 2.1. Materials

MWCNTs were obtained from LG Chem, Korea. Sulfuric acid and nitric acid (Daejungchem, Korea), ethylenediamine (Sigma Aldrich, Seoul, Korea), HATU(1-[Bis(dimethylamino)methylene]-1H-1,2,3-triazolo[4–b]pyridinium 3-Oxide Hexafluorophosphate) (Daejungchem, Shihsing, Korea), and PMMA (M.W. 35,000, Alfa Aesar, Seoul, Korea) were used for surface modification of MWCNTs. As-obtained MWCNTs, PMMA, and PS (M.W. 25,000, Sigma Aldrich, Seoul, Korea) were used for fabricating composites.

### 2.2. Surface Modification of MWCNTs

The MWCNTs were purified and functionalized with carboxylic acid functional groups as follows; 4 g of MWCNTs were dispersed in a mixture of nitric acid (100 mL) and sulfuric acid (300 mL) and sonicated for 2 h at room temperature to form a suspension [12]. After sonication, suspension was heated to 80 °C and stirred for 1 hour. Acid treatment produces carboxylic acid functional groups on the surface of MWCNTs and eliminated impurities. Then, the suspension was washed with excess DI water using centrifugation and vacuum filtration until its pH was 6. Lastly, MWCNT–COOH were dried in a convection oven overnight.

Half a gram of MWCNT–COOH was dispersed in 250 mL of ethylenediamine using an ultrasonicator until it formed a homogeneous suspension. After sonication, 25 mg of HATU(*N*-[(dimethylamino)-1H-1,2,3-triazolo[4–6]pyridin1-ylmethylene]-*N*-methylmethanaminium hexafluorophosphate *N*-oxide), which is used as a coupling agent, was added and sonicated for 4 h at 40 °C [13]. The product was washed with excess DI water using vacuum filtration for 3 times. MWCNTs–NH_2_ was dried in a convection oven for 4 h at 80 °C.

Half a gram of MWCNTs–NH_2_ was dispersed in THF using an ultrasonicator, while 10 g of PMMA was separately dispersed in THF by magnetic stirring at 60 °C. Completely dispersed MWCNTs–NH_2_ were mixed with PMMA solution and sonicated to form homogeneous mixture for 6H. The mixture was magnetic stirred for 24 h at 60 °C and then cooled to room temperature. Finally, it was washed with excess THF using vacuum filtration for 5 times to remove unreacted PMMA [14].

### 2.3. Fabrication of PMMA/PS/MWCNT

Solution mixing was chosen to fabricate PMMA/PS blends and PMMA/PS/CNT composites [15]. The total mass of the batch was constrained to 1.5 g. The concentrations of MWCNTs and surface treated MWCNTs are 0 wt%, 0.5 wt%, 1 wt%, 3 wt%, and 5 wt% for each composite. PMMA and PS are dissolved in THF (tetrahydrofuran), which has a good solubility for both polymers [16]. Then MWCNTs are dispersed in THF separately by probe sonication for 20 min under 750 W and 20 kHz. After that, MWCNT suspension was poured into polymer solution. The solution was stirred and heated on a hot plate until solvent almost evaporated. Then viscous solution was dried at room temperature overnight. All the composites are fabricated in the same way.

### 2.4. Characterization

The functionalized MWCNTs were characterized by X-ray photoelectron spectroscopy (XPS, Thermo UK Kalpha, Seoul, Korea) using an Al Kalpha X-ray source (1486.6 eV) and a hemispherical analyzer. Fourier-transform infrared spectroscopy (FT-IR; Perkin-Elmer Inc. Spectrum One, Seoul, Korea) was used to investigate the surface modification of MWCNTs. During curve fitting, the Gaussian peak widths were constant in each spectrum. Probe sonicator (VCX 750, Sonics & Materials, Inc, Newtown, Connecticut, USA) was used to disperse MWCNTs in solvent for fabricating composites. Thermogravimetric analyses (TGA; TGA-2050, TA Instruments, New Castle, DE, USA) of the samples were used to perform the thermal decomposition process. Two milligram samples were heated to 800 °C at a heating rate of 20 °C·min^−1^ under a nitrogen atmosphere. A Cryo Ultramicrotome (PT PC Ultramicrotome & Phoographic) was used to make samples for HR-TEM. High-resolution transmission electron microscope (HR-TEM, JEM-3010, JEOL Ltd. Tokyo, Japan) and field-emission scanning electron microscopy (FE-SEM, Sigma, Carl Zeiss, Oberkochen, Germany) were used to observe the morphology of the surface modified MWCNTs and cross-sections of composites. The thermal conductive performance of the composites was characterized by laser flash analysis (LFA, Netzsch Instruments Co, Nanoflash LFA447, Selb, Bavaria, Germany) at room temperature. The bulk densities (g·cm^−3^) of the specimens were measured using the Archimedes water displacement method. Thermal conductivity (*k*) was calculated by multiplying the thermal diffusivity, density, and specific heat capacity of the composite.

Solvent extraction study was used to evaluate the degree of the continuous polymer phase. Blend samples were dipped in 10 mL of cyclohexane for 6h to dissolve the PS phase.

## 3. Results and Discussion

### 3.1. Infrared Spectroscopy

Figure 1 shows FT-IR spectra of pristine MWCNT and surface modified MWCNTs. The reason of broad band at 3400–3500 cm^−1^ is presence of O–H groups on the surface of pristine MWCNTs and is believed to result from either moisture in the air being tightly bound to the MWCNTs or reaction with oxygen during refinement of the pristine MWCNTs. The peak at 1650 cm^−1^ is due to alkenyl C=C stretch. Unsaturated C=C bonds are produced because of O–H groups. In the IR spectrum of oxidized MWCNT, the reason for the peak at 1725 cm^−1^ is the C=O stretch of the carboxylic group [14]. The IR spectrum of amide-functionalized MWCNTs shows the removal of the band at 1725 cm^−1^ and formation of a band at 1660 cm^−1^ because of amide carbonyl (C=O) stretching. In addition, the presence of new band at 1214 cm^−1^ is attributed to C–N bond stretching, corroborating amine functional group [13]. The IR spectrum of PMMA-grafted MWCNT shows slightly clearer peaks than that of amide-functionalized MWCNTs at 1660 cm^−1^ that represents amide carbonyl(C=O) stretching because of adding amide groups. Also, sharp peaks at 1400 cm^−1^, 1370 cm^−1^, and 1310 cm^−1^ represent alkane (C–H) bending caused by grafted PMMA. Peaks at 1200 cm^−1^ and 1134 cm^−1^ represented ester (O=C–O) bond stretching in the grafted PMMA.

### 3.2. X-ray Photoelectron Spectroscopy

The XPS spectra for pristine and functionalized MWCNTs are shown in Figure 2. Samples were dried in a convection oven overnight prior to analysis. Figure 2 shows spectra of pristine MWCNT, MWCNT–COOH, MWCNT–NH_2_, and MWCNT–PMMA. Peaks at 284.7 eV and 285.2 eV indicate carbon atoms bonded to other functional groups. It can be found that pristine MWCNT contains carbon atoms and trace oxygen atoms through 286.2 eV that might be affected by atmosphere or by purification. After acid treatment, the contents of oxygen atoms increase from 0.42% to 10.73%. The additional binding energies at 286.5 eV and 288.9 eV shown in Figure 2b represent the –C–OH and O–C=O (carboxylic acid) groups, respectively [17,18]. For the amide functionalized MWCNTs shown in Figure 2c, the XPS results show stronger binding energies at 286.2 eV and 288.2 eV that represent C–N and HN–C=O respectively. After amide functionalization, the contents of nitrogen atoms increased to 3.37%. It confirms conversion of carboxylic acid group to amide group. In XPS spectrum of PMMA–*g*–MWCNT, 288 eV which represent binding energy of HN-C=O that connect PMMA and MWCNT–NH_2_; 289 eV indicates carbon in ester functional group; 286.7 eV represents carbon in methyl group that connect with ester group; binding energies at 289 eV and 286.7 eV are included in PMMA-grafted onto the surface of MWCNT.

### 3.3. Morphology(MWCNT-g-PMMA)

Figure 3 and Figure 4 show the morphology of pristine MWCNT and PMMA–*g*–MWCNT. PMMA was grafted onto the surface of as-received MWCNT. The diameters of PMMA–*g*–MWCNT are ~50 nm; on the other hand, those of pristine MWCNT are ~20 nm. It is 2–3 times larger after PMMA grafting. Amino-functionalized carbon nanotube has highly reactive amine functional groups. These functional groups act as reaction site to graft PMMA. Because PMMA is bigger than other functional group like COOH or ethylenediamine, the diameter of sites where PMMA was attached increased. Also, PMMA usually has somewhat different molecular weight, which makes uneven surface. These made MWCNT-*g*-PMMA have rougher surface after PMMA grafting.

Figure 5 shows a HR-TEM microphotographs of fabricated composites. Upper ones included 10 wt% PS, 90 wt% PMMA, and 0.5 wt% pristine MWCNTs. The other ones included 10 wt% PS, 90 wt% PMMA, and 0.5 wt% PMMA-grafted MWCNTs. Comparing these pictures, composites with PMMA-grafted MWCNTs formed more efficient routes for phonon transfer. Owing to polymer grafting, PMMA-grafted MWCNTs were localized in PMMA phase more than pristine MWCNTs are. When not using polymer grafting, MWCNTs were dispersed in polymer blends at some parts. This phenomenon occurred because affinity between a filler and a matrix enhanced. After grafting, molecular structure of filler and a matrix turned into similar structure. Also, polymer grafting contributed to enhance the dispersion state of filler that influence thermal conductivity. Poor dispersion state could cause a decrease in phonon mean free path leading to structure boundary scattering and defect scattering, which means poor thermal conductivity [19].

### 3.4. TGA(MWCNT–PMMA)

To figure out the composition of the functionalized group in MWCNT-*g*-PMMA, TGA was used at a heating rate of 20°C/min from 25 °C to 800 °C under a N_2_ to investigate respective stage of functionalized MWCNTs. Under these conditions, weight loss approximately 15%, 21%, 26%, 70%, and 100% were observed for raw MWCNT, MWCNT–COOH, MWCNT–NH_2_, and MWCNT-*g*-PMMA, and raw PMMA, respectively, because of thermal degradation of the functionalized group grafted onto the surface of MWCNT. TGA data in Figure 6 and an equation were used to figure out the degree of functionalization:
*Func. CNT = (moles of functional group/moles of carbon in CNTs)*
According to the equation, 1.7%, 1%, and 0.00857% were acquired for acid functionalization, amino functionalization, and PMMA grafting, respectively [20].

### 3.5. SEM (Composite)

Field-emission scanning electron microscopy (FE-SEM, Sigma, Carl Zeiss) was used to observe the morphology of samples. Figure 7 shows a cross-section of PS/PMMA blends. Phase separation between two polymers is found because of their immiscibility. The morphology of composites varies depending on the composition of PS and PMMA. SEM image of PS/PMMA(90/10) has domain–matrix system. Dark site and bright site represents PMMA and PS respectively. In the SEM image, PMMA forms random shape and PS occupies spacious background. PMMA is assigned as a minor phase because of its low composition. SEM image of PS/PMMA(50/50) shows somewhat different micrograph. It shows co-continuous structure. Figure 7c,d looks stained because of phase separation. The PMMA phase is much larger than that of picture, which contains 10wt% of PMMA. SEM image of PS/PMMA(10/90) has structure which is the opposite with composites that contain 90wt% of PS. PMMA is seemingly major part in the composite, which make darker image than other structures. Figure 8a,b showed an SEM image of PMMA/PS(50/50) blend after etching of the PS phase(using cyclohexane). Co-continuous structure was figured out again after PS phases were dissolved in cyclohexane as shown in Figure 7.

Figure 9 shows samples with differences in MWCNT concentration. This paper focus on composites that contain 90 wt% of PMMA because PMMA is assigned as a matrix, which can enhance thermal conductivity. MWCNTs are selectively localized in PMMA phase around PS domain and interface of between two immiscible polymers. This phenomenon occurs because of the polar nature that belongs to PMMA phase and MWCNTs [21]. PS domain forms circular shape when not containing MWCNTs. Composites that contain MWCNTs lose their circular shape and form line shape because MWCNTs are surrounding PS domain and fill the matrix. When this line is formed, size of PS domain decreases. It is because viscosity of PMMA phase increases, due to input of MWCNTs [8]. This increment of viscosity makes it hard for PS domain to maintain their spherical shape and size when samples didn’t contain MWCNTs.

### 3.6. LFA (Composite)

The thermal conductivity of fabricated composites is the most important data of this paper. This equation is used to calculate thermal conductivity (*k*).
*K = α ρ Cp*
In the equation, *ρ* is sample density, *Cp* is heat capacity of sample and *α* is thermal diffusivity of sample. Figure 10 shows the thermal conductivity of samples at 25 °C. 

Figure 10 shows the thermal conductivity of PMMA/CNT, PS/CNT, PMMA/PS/PMMA–*g*–CNT, and PMMA/PS/CNT composites as a function of filler loading. This data reveals the effect of using double percolation system and surface modification of filler. In PMMA/CNT and PS/CNT, MWCNTs are homogeneously dispersed in matrix. However, using two immiscible polymers, there was a tendency for MWCNTs to be situated in PMMA phase [15]. It contributed to form a filler network efficiently at the same filler loading. In the Figure 10, all the samples contain 10 wt% PS and 90wt% PMMA. In this case, PMMA functions as a matrix that contain filler. 

Thermal conductivity of composites with surface functionalized MWCNTs was higher than that of composites with pristine MWCNTs at same filler loading [22]. The effectiveness of polymer grafting was increasing as filler concentration increased. PMMA grafting resulted in relatively higher interfacial affinity between carbon nanotubes and PMMA phase. The affinity between a particle and a matrix was correlated to the resemblance of molecular shape. Resemble molecular structure between a filler and a matrix leaded to a facile incorporation. This enhanced affinity reduced both the interfacial thermal resistance and phonon scattering. Also, improved affinity affects localization state of composites [23]. When wrapped by PMMA, MWCNTs are more likely to localize in PMMA as observed in TEM micrograph; it is the reason for the formation of an efficient phonon transfer route. These factors accorded with the Figure 10 that showed increasing trend in thermal conductivity curve.

### 3.7. IR Camera (Composite)

Figure 11 shows some photos of IR camera. Color goes brighter with temperature. IR pictures of PS/MWCNT, PMMA/MWCNT, and PS/PMMA/MWCNT (10/90/3wt%) are arranged in a row from left to right. All samples contain 3wt% of MWCNTs. Color of samples goes bright as temperature goes up. In Figure 10, sample on right one has the brightest color of three samples. These color changes revealed the effect of phase separation using two polymers.

In addition, Figure 12 shows the effect of polymer grafting; the left one includes pristine MWCNTs and the right one includes PMMA-grafted MWCNTs. The right one has a brighter color in all of the pictures. The contrast of color between two samples has the effect of polymer grafting onto the surface of filler.

## 4. Conclusions

In this paper, two immiscible polymers and MWCNTs were used for fabricating composites. When using two polymers and surface modification, thermal conductivity was enhanced by approximately 15%, 27%, and 30% at 0.5 wt%, 1 wt%, and 3 wt% of MWCNT in PMMA matrix, respectively. In addition, 36%, 39%, and 82% of thermal conductivity improvement was accomplished at 0.5 wt%, 1 wt%, and 3 wt%, respectively, compared with using PS for matrix and MWCNT for filler. Grafted PMMA enhances affinity between PMMA phase in matrix and filler: it helps form the phonon transfer route and selectively localizes fillers in the PMMA phase. These two reasons contributed thermal conductivity.

## Figures and Tables

**Figure 1 polymers-11-01347-f001:**
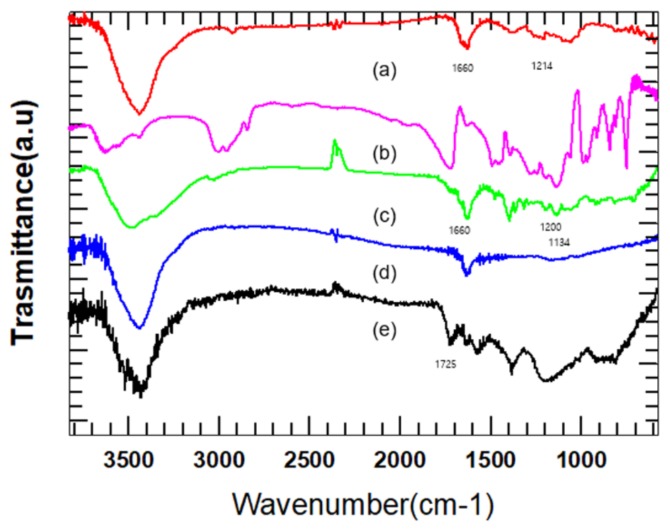
FT-IR spectra of multiwalled carbon nanotubes (MWCNTs): (**a**) MWCNT–NH_2_, (**b**) PMMA, (**c**) PMMA–*g*–MWCNT, (**d**) raw MWCNT, and (**e**) MWCNT–COOH.

**Figure 2 polymers-11-01347-f002:**
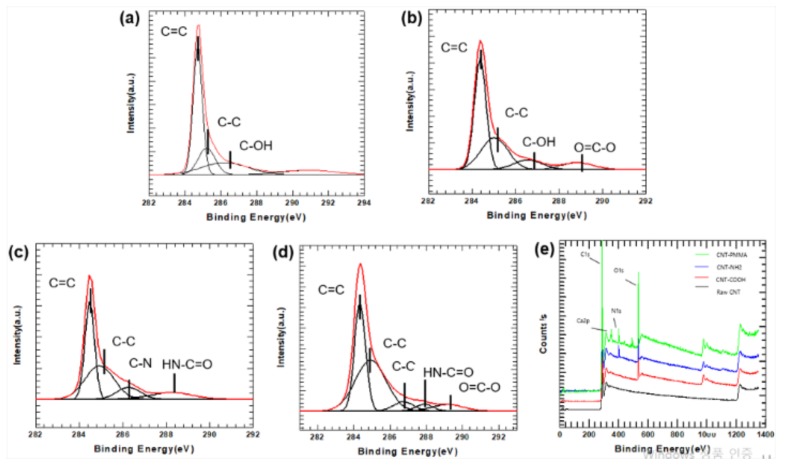
X-ray photoelectron spectroscopy (XPS) spectra of MWCNTs: (**a**) pristine MWCNT, (**b**) MWCNT–COOH, (**c**) MWCNT–NH_2_, and (**d**) PMMA–*g*–MWCNT; (**e**) wide scan spectra.

**Figure 3 polymers-11-01347-f003:**
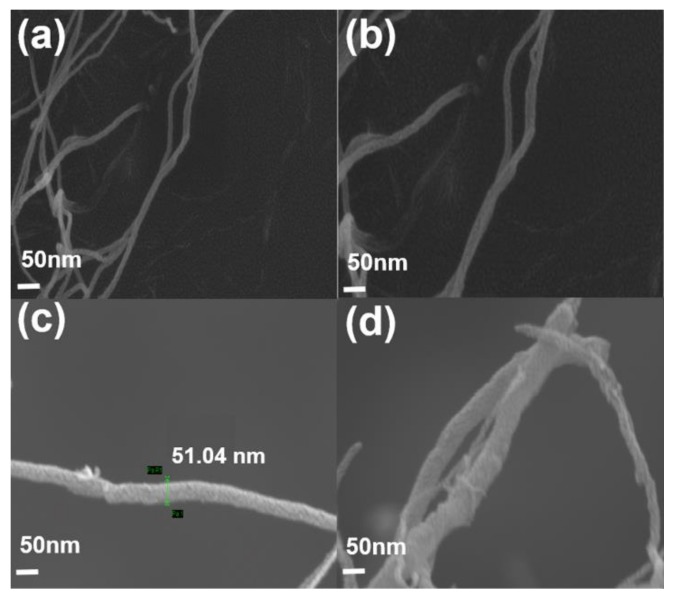
The SEM image of MWCNTs: (**a**,**b**) pristine MWCNT and (**c**,**d**) PMMA–*g*–MWCNT.

**Figure 4 polymers-11-01347-f004:**
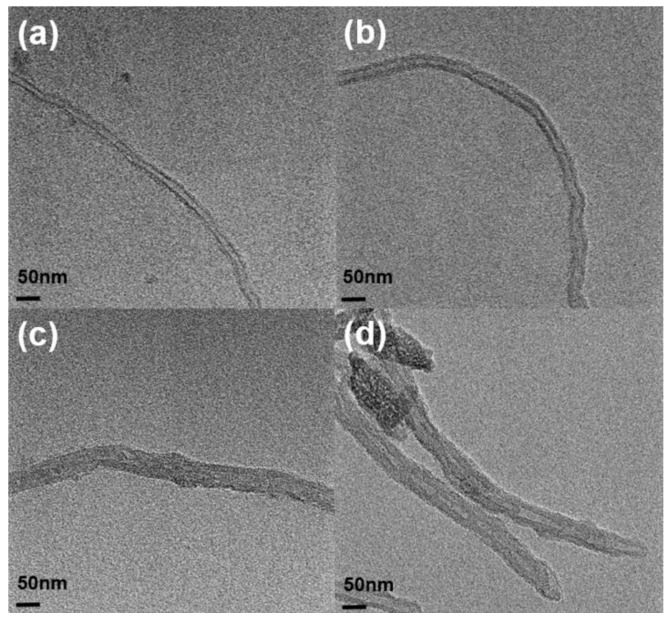
The TEM image of MWCNTs: (**a**,**b**) pristine MWCNT and (**c**,**d**) PMMA–*g*–MWCNT.

**Figure 5 polymers-11-01347-f005:**
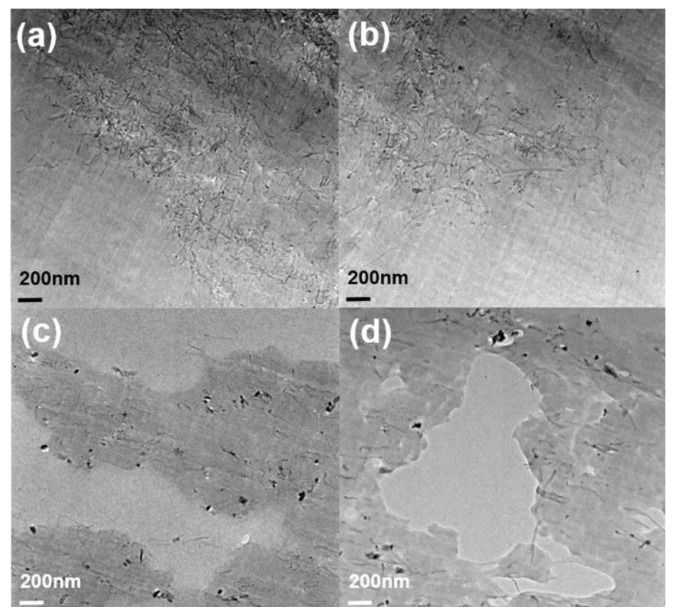
TEM image of PMMA/PS/CNT(90/10/0.5wt%) composites: (**a**,**b**) PMMA/PS/CNT and (**c**,**d**) PMMA/PS/PMMA–*g*–CNT.

**Figure 6 polymers-11-01347-f006:**
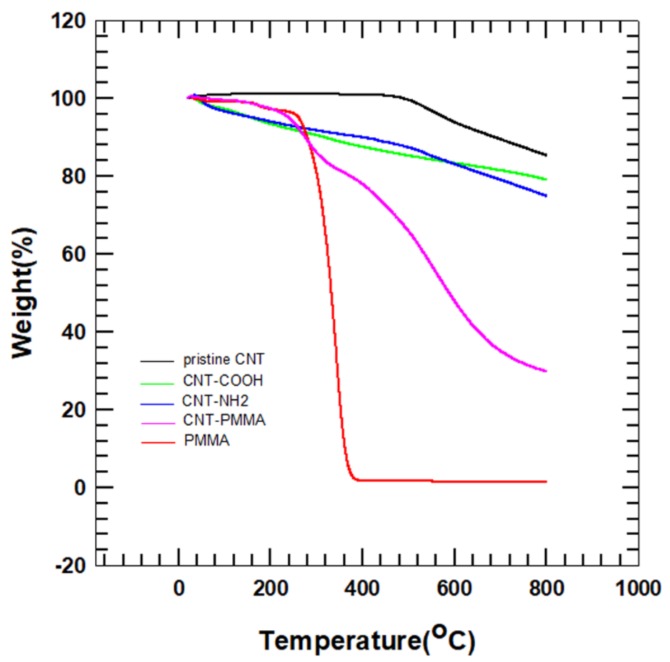
The thermogravimetric analysis (TGA) image of pristine MWCNTs and PMMA-grafted MWCNTs.

**Figure 7 polymers-11-01347-f007:**
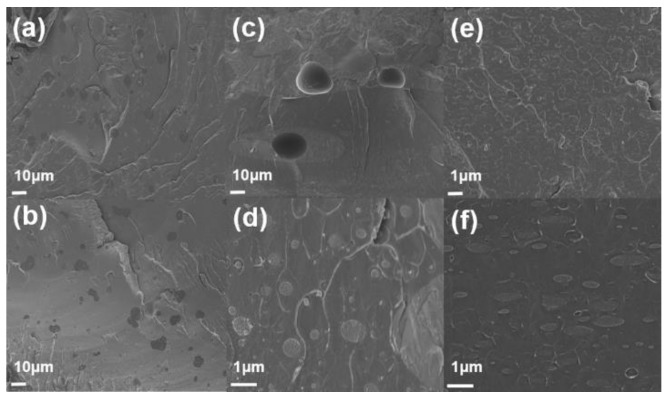
The SEM cross-section images of the PMMA/PS blends: (**a**,**b**) PMMA/PS(10/90), (**c**,**d**) PMMA/PS(50/50), and (**e**,**f**) PMMA/PS(90/10).

**Figure 8 polymers-11-01347-f008:**
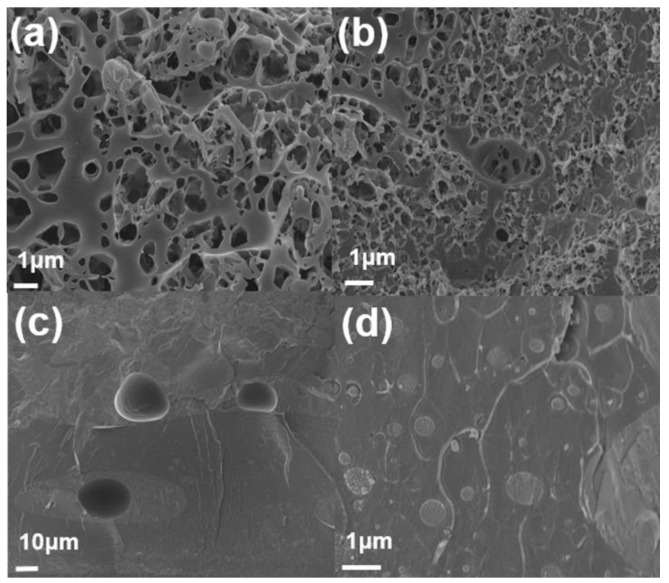
The SEM cross-section images of the PMMA/PS(50/50) blends. (**a**,**b**) PS phase-etched blend PMMA/PS(50/50) and (**c**,**d**) PMMA/PS(50/50) without etching.

**Figure 9 polymers-11-01347-f009:**
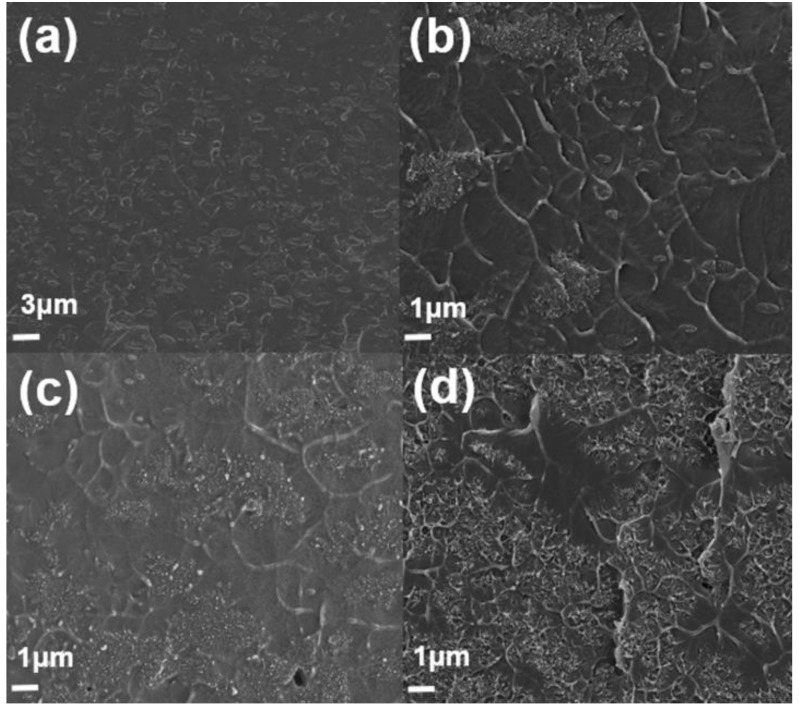
The SEM image of PMMA/PS/CNT composites. (**a**) PMMA/PS(90/10), (**b**) PMMA/PS/CNT(90/10/0.5wt%), (**c**) PMMA/PS/CNT(90/10/1wt%), and (**d**) PMMA/PS/CNT(90/10/3wt%).

**Figure 10 polymers-11-01347-f010:**
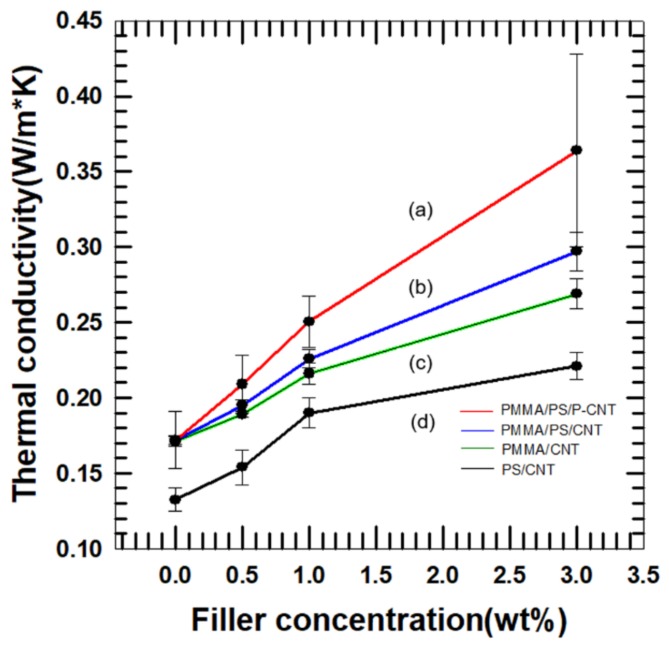
Thermal conductivity of composites. (**a**) PMMA/PS/PMMA–*g*–CNT, (**b**) PMMA/PS/CNT, (**c**) PMMA/CNT, and (**d**) PS/CNT.

**Figure 11 polymers-11-01347-f011:**
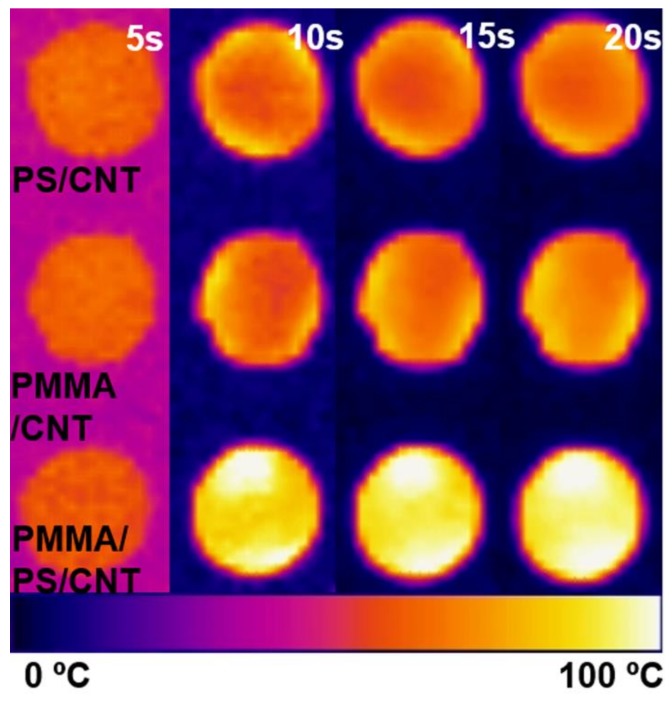
IR image of samples that represent the effect of double percolation.

**Figure 12 polymers-11-01347-f012:**
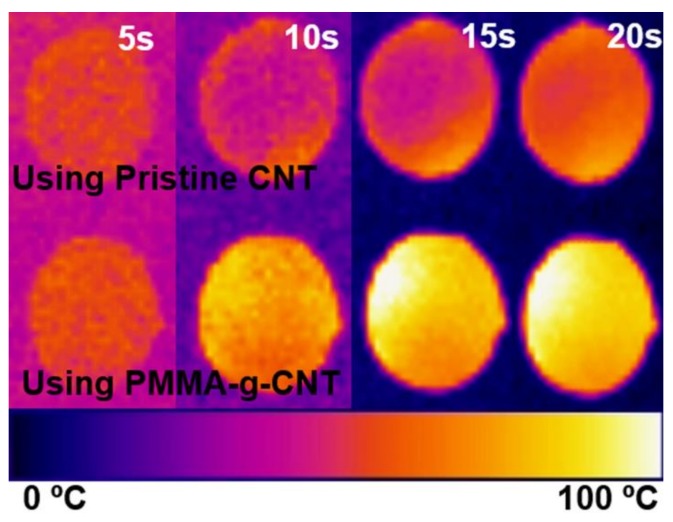
IR image of samples that represent the effect of polymer grafting on the surface of MWCNT.

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
