# Peer review of "Thermal Conductivity Enhancement Derived from Poly(Methyl Methacrylate)-Grafted Carbon Nanotubes in Poly(Methyl Methacrylate)/Polystyrene Blends"

_polymers, 2019, doi:10.3390/polym11081347_

Round 1

Reviewer 1 Report

This paper deals with the improvement in thermal conductivity of ternary polymer composites made of PMMA/PS blend and multi walled carbon nanotubes (MWCNTs).

The topic of this manuscript is of current interest since new thermally conductive polymer composites are in high demand for several applications. However, to be published in Polymers journal this paper needs major revisions. First, the reviewer suggests a complete rewriting of the manuscript in order to organize better the discussion of the results. The authors produced and analyzed in their study many systems:

·         blends PMMA/PS with different relative amounts of the polymeric phases;

·         many hybrid blends with several contents of MWCNTs, both pristine and with different surface treatments.

However, in the reviewer opinion the obtained results were not well discussed and correlated each other. It would be better to analyze fewer systems, but more in detail, and not to report the data according to the characterization techniques used. It would be more appropriate to highlight the effectiveness of the different CNT surface modifications and the effects of the several variables analyzed (such as the relative amounts of PMMA and PS, the MWCNT content and surface modification) on the morphology and thermal conductivity of the polymer composites.

In the following, some specific remarks are reported.

The reviewer suggests a careful reading of the paper in order to correct some grammatical errors that appear in the manuscript; for example at page 2 line 43 the authors has to substitute the words “they enhances” with “they enhance”.

In the paragraph “Characterization” of the Experimental section, the authors must also report the FTIR technique.

At page 5 line 148 the authors refer to Figure X, but they probably intend Figure 2.

The reviewer suggests avoiding paragraphs with only few lines and a figure, such as paragraphs 3.3, 3.4 (this paragraph is reported twice) and 3.5. Moreover, in most cases the authors just comment the figures, without a real interpretation of the obtained results.

At page 6 line 174 the authors write “MWCNT-g-PMMA have rougher surface before PPMA grafting”, probably they intend “MWCNT-g-PMMA have rougher surface after PPMA grafting”.

The reviewer suggests rewriting the comments on TGA curves (Figure 5) for pristine and PMMA-g-MWCNTs samples (at page 6 lines 178-181). In particular, the authors write: “as received MWCNT…shows almost 0% of weight loss until temperature went 800°C”, but in Figure 5 a weight loss of about 15% can be observed. Moreover, regarding the MWCNT-g-PMMA sample, a weight loss of about 70% is actually shown, but this weight loss has to take into account not only the grafted PMMA, but also the material loss observed for the pristine MWCNT.

In Figure 6 the micrographs (e) and (f) probably refer to the blend PMMA/PS (90/10) and not to the blend PMMA/PS (10/90). However, a selective etching of one of the polymeric phases in the blends PMMA/PS would be useful to evidence better the differences in the blends morphologies.

In the legend of Figure 7, regarding the SEM image (b), the authors probably refer to the sample PMMA/PS/CNT (90/10/0.5%).

At page 8 line 229 and at page 9 line 236 the authors should report the numbers of the Figures they refer.

Significant experimental errors may be involved in thermal conductivity measurements, particularly when indirect methods are carried out, such as laser flash analysis used in the present research. In other words, since the accuracy of thermal conductivity measurements are typically in the range of 5–10% [1], and the improvement obtained in this study is of about 13%, it would be opportune to report in the paper the standard deviations of the thermal conductivity measurements performed, in order to correctly evaluate the obtained results.

[1] Zhidong Hana, Alberto Fina, Thermal conductivity of carbon nanotubes and their polymer nanocomposites: A review, Progress in Polymer Science 36 (2011) 914–944

Author Response

Thank you for your sincere comments on this paper.

We have examined each point and revised the manuscript accordingly.

Please see the attachment​

Reviewer 2 Report

The present manuscript describes the thermal conductivity properties of PS/PMMA blends in the presence of MWCNTs with or without surface functionalization with PMMA grafted chains. 

Although the manuscript could have a potential interest to the scientific community, its present form is not sufficient for publication at this stage. 

Significant language and grammar editing is necessary. Several experimental details are missing. An annotaded pdf file of the submited manuscript with comments has been uploaded to possibly help the authors correct and resraft their manuscript.

Author Response

Thank you for your sincere comments on this paper.

We have examined each point and revised the manuscript accordingly.

Reviewer’s comments: The present manuscript describes the thermal conductivity properties of PS/PMMA blends in the presence of MWCNTs with or without surface functionalization with PMMA grafted chains. 

Although the manuscript could have a potential interest to the scientific community, its present form is not sufficient for publication at this stage. 

Significant language and grammar editing is necessary. Several experimental details are missing. An annotaded pdf file of the submited manuscript with comments has been uploaded to possibly help the authors correct and resraft their manuscript.

Point1: reviewer confused these sentences

By analyzing morphology of composites, the composition of PS and PMMA that assigned PMMA to matrix phase was revealed. After that, to help MWCNTs localized in PMMA phase, polymer grafting was performed. Finally, the thermal conductivity of PS/PMMA/PMMA-g-MWCNT composites was investigated.

Response1: author changed those three sentences like below for you to understand

To help MWCNTs localized in PMMA phase and enhancement of dispersion state of MWCNTs, polymer grafting was performed. Finally, the thermal conductivity of composites was investigated.

Point2: reviewer wanted to include characteristics of the commercial PMMA like MWs

Response2: average molecular weight of PMMA is 35000g/gmol.

Point3: reviewer wanted to know ‘which probe sonicator was used and under what conditions’

Response3: Probe sonicator is VCX 750 made by Sonics & Materials, Inc. Author used it under 750W and 20Khz to disperse MWCNTs in solvent.

Point4: Author wanted to know information about FT-IR instrumentation

Response4: Fourier-transform infrared spectroscopy(FT-IR; Perkin-Elmer Inc. Spectrum One) were used to investigate the surface modification of MWCNTs.

Point5: Author suggested that auhor add and compare the PMMA FT-IR spectrum. Possibly include magnified areas of interest. Several of the noticed peaks have been assigned in the literature to different groups please revise and/or explain thoroughly.

Response5: Author added PMMA FT-IR in the figure 1. In addition, author added digits on the figure1 to apprehend numerical value. Author used FT-IR spectrum table of sigma-aldrich to investigate FT-IR data to investigate IR spectrum of PMMA-g-CNT. Author attached the link of web site. Also, author used the literature to investigate IR data..

reference

https://www.sigmaaldrich.com/technical-documents/articles/biology/ir-spectrum-table.html

Ramanathan, T., Fisher, F. T., Ruoff, R. S., & Brinson, L. C, Amino-functionalized carbon nanotubes for binding to polymers and biological systems. Chemistry of Materials, 2005,17(6), 1290-1295.

Point6: Reviewer suggested that author include the whole spectra range namely the “Wide-scan spectra”

Response6: Author added survey of XPS data in the figure 2 .

Point7: Reviewer pointed that how to measure or estimate rougher surface of MWCNT-g-PMMA

Response7: Amino functionalized carbon nanotube has highly reactive amine functional groups. These functional groups act as reaction site to graft PMMA. Because PMMA is bigger than other functional group like COOH or ethylenediamine, the diameter of sites where PMMA was attached increased. Also, PMMA usually has somewhat different molecular weight, which makes uneven surface. These made MWCNT-g-PMMA have rougher surface after PMMA grafting. Expression ‘rough’ surface is extracted in reference.

reference

Choi, E. Y., Nam, J. U., Hong, S. H., & Kim, C. K. (2018). Characteristics of Polycarbonate Composites with Poly (methyl methacrylate) Grafted Multi-Walled Carbon Nanotubes. Macromolecular Research, 26(2), 107-112.

Point8: Reviewer suggested that author add pure PMMA, CNT-COOH, CNT-NH2. Also, Reviewer pointed that 800oC MWCNTs show an ~15wt% loss and the functionalization degree of functionalized carbon nanotubes.

Response8: Author added TGA data of pure PMMA, CNT-COOH and CNT-NH2. Also, author attatched reference for TGA data of raw CNT from other literature. Author estimated that impurities of CNT and defect in CNT made weight loss as shown figure5. To calculate the accurate functionalization degree of the grafted MWCNTs, author used the equation: The fraction of carbon atoms that are functionalized with PMMA can be calculated with the following equation: (0.20/35000)/(0.80/12)) = 0.00857% for PMMA grafting

(0.06/45)/(0.94/12)=1.7% for acid functionalization..

(0.05/60)/(0.95/12)=1% for amino functionalization.

Also, 15wt% or more weight loss of raw carbon nanotube is found in other literature.

reference

Chungui Zhao, Lijun Ji, Huiju Liu, Guangjun Hu, Shimin Zhang, Mingshu Yang, Zhenzhong Yang, functionalized carbon nanotubes containing isocyanate groups, journal of solid state chemistry, 2004, 177, 4394-4398.

Yuan-Pin Huang, I-Jou Lin, Chih-Chen Chen, Yi-Chiang Hsu, Chi-Chang Chang and Mon-Juan Lee, Delivery of small interfering RNAs in human cervical cancer cells by polyethylenimine-functionalized carbon nanotubes, nanoscale research, 2013, 8, 267

Point 9: Reviewer required to add reference on sentences ‘The system has especially received attention from people who were researching electric conductivity. Recently some people have brought this system for thermal conductivity.’

Response 9: Author added reference on sentences. Paragraph 8.3.4 said that people in thermal conductivity only recently have interest in double percolation system.

reference

Chen, V.V. Ginzburg, J. Yang, Y. Yang, W. Liu, Y. Huang, L. Du, B. Chen, Thermal conductivity of polymer based composites: Fundamentals and applications, Prog Polym Sci. 2016, 59, 41-85.

Point 10: Reviewer required to add reference on sentences ‘If filler was situated in matrix phase, the

improvement of thermal conductivity was maximized’

Response10: In double percolation system, ratio of incorporated polymer decide a role for matrix or domain. It is easier for filler to form heat transfer route in matrix because filler is easily connected to each other. Reference [1] said polymer that occupy large portion in blend act as a continuous phase. Reference [2] said selective localization in continuous phase helps to achieve high TC.

reference

[1] Poothanari, M. A., Abraham, J., Kalarikkal, N., & Thomas, S. (2018). Excellent electromagnetic interference shielding and high electrical conductivity of compatibilized polycarbonate/polypropylene carbon nanotube blend nanocomposites. Industrial & Engineering Chemistry Research, 57(12), 4287-4297.

[2] H. Chen, V.V. Ginzburg, J. Yang, Y. Yang, W. Liu, Y. Huang, L. Du, B. Chen, Thermal conductivity of polymer based composites: Fundamentals and applications, Prog Polym Sci. 2016, 59, 41-85.

We hope that this revision addresses the reviewers’ comments.

Sincerely

Round 2

Reviewer 1 Report

The authors have essentially made all the suggested corrections.